# Simultaneous Promotion of Salt Tolerance and Phenolic Acid Biosynthesis in *Salvia miltiorrhiza* via Overexpression of *Arabidopsis MYB12*

**DOI:** 10.3390/ijms242115506

**Published:** 2023-10-24

**Authors:** Tianyu Li, Shuangshuang Zhang, Yidan Li, Lipeng Zhang, Wenqin Song, Chengbin Chen, Weibin Ruan

**Affiliations:** College of Life Sciences, Nankai University, Tianjin 300071, China; litianyu582@163.com (T.L.); ylnmll99@163.com (S.Z.); danicali2021@163.com (Y.L.); nknanhai@163.com (L.Z.); songwq@nankai.edu.cn (W.S.)

**Keywords:** *Salvia miltiorrhiza*, *AtMYB12*, salt tolerance, phenolic acid

## Abstract

Transcription factors play crucial roles in regulating plant abiotic stress responses and physiological metabolic processes, which can be used for plant molecular breeding. In this study, an R2R3-MYB transcription factor gene, *AtMYB12*, was isolated from *Arabidopsis thaliana* and introduced into *Salvia miltiorrhiza* under the regulation of the CaMV35S promoter. The ectopic expression of *AtMYB12* resulted in improved salt tolerance in *S. miltiorrhiza*; transgenic plants showed a more resistant phenotype under high-salinity conditions. Physiological experiments showed that transgenic plants exhibited higher chlorophyll contents, and decreased electrolyte leakage and O_2_^−^ and H_2_O_2_ accumulation when subjected to salt stress. Moreover, the activity of reactive oxygen species (ROS)-scavenging enzymes was enhanced in *S. miltiorrhiza* via the overexpression of *AtMYB12*, and transgenic plants showed higher superoxide dismutase (SOD), catalase (CAT), and peroxidase (POD) activities compared with those of the wild type (WT) under salt stress, coupled with lower malondialdehyde (MDA) levels. In addition, the amount of salvianolic acid B was significantly elevated in all *AtMYB12* transgenic hair roots and transgenic plants, and qRT-PCR analysis revealed that most genes in the phenolic acid biosynthetic pathway were up-regulated. In conclusion, these results demonstrated that *AtMYB12* can significantly improve the resistance of plants to salt stress and promote the biosynthesis of phenolic acids by regulating genes involved in the biosynthetic pathway.

## 1. Introduction

*Salvia miltiorrhiza* Bunge, known as Danshen, belongs to the *Labiatae* family. Its red storage roots are mainly used for the treatment of cardiovascular and cerebrovascular diseases [1,2]. The active pharmaceutical ingredients of Danshen include two major groups: lipophilic tanshinones and hydrophilic phenolic acids [3]. As the main active ingredient of phenolic acids, salvianolic acid B (SalB) plays an important role in antioxidant and free radical scavenging [4]. Phenolic acids are primarily synthesized through the phenylpropanoid- and tyrosine-derived branch pathway, and most biosynthetic genes encoding enzymes including tyrosine aminotransferase (TAT), hydroxyphenyl pyruvate reductase (HPPR), phenylalanine ammonia-lyase (PAL), cinnamate-4-hydroxylase (C4H), 4-coumarate-CoA ligase (4CL), rosmarinic acid synthase (RAS), and a cytochrome P450-dependent monooxygenase (CYP98A14) have been characterized and cloned [5,6,7,8,9,10]. The metabolic engineering of these genes individually or simultaneously has been utilized to enhance phenolic acid accumulation [9,11].

Abiotic stress leads to an excessive accumulation of ROS, disrupting the metabolic balance of cells and affecting plant growth. To cope with oxidative stress, plants have evolved complex efficient antioxidant mechanisms to quickly adapt to the condition. Natural plant antioxidant enzymes and non-enzymatic antioxidants can mitigate the harmful effects of different types of abiotic stress, including drought, salinity, and fungicide [12,13,14]. Salinity is a widely existing form of abiotic stress that affects plant growth and productivity. To date, there is evidence that a number of transcription factors, including WRKY, NAC, and the MYB family, are involved in regulating the response to salt stress [15,16,17]. In contrast to functional proteins, a single transcription factor can regulate a set of downstream target genes and then regulate the physiological and biochemical processes of plants to cope with salt stress. Thus, transcription factors can serve as excellent candidate genes for salt tolerance and have great development prospects in plant molecular breeding.

MYB proteins comprise a huge family of TFs that play important roles in plant development and secondary metabolism. According to the number of repetitions of conserved domains on the N-terminal, MYB TFs are classified into four subfamilies: 1R-MYBs, R2R3-MYBs, R1R2R3-MYBs, and 4R-MYBs. The majority of MYB proteins belong to the R2R3-MYB subfamily, and members of this subfamily have been reported to be involved in the regulation of abiotic stress [18,19,20,21]. For instance, *ThMYB8*, which belongs to the R2R3-MYB subfamily, can significantly affect the seed germination rate and root growth of *Tamarix hispida* under salt stress [22]. Furthermore, the overexpression of *OsMYB2* greatly conferred tolerance to rice against salt, cold, and dehydration stress [23]. *GhMYB73*, a MYB gene which was isolated from *Gossypium hirsutum*, and its overexpression in *Arabidopsis thaliana* significantly increase tolerance to salt stress [24]. Meanwhile, some research showed that R2R3-MYB TFs are involved in the regulation of phenolic acid biosynthesis in *Salvia miltiorrhiza*. Typically, *SmMYB98* could regulate the biosynthesis of phenolic acids positively [25]. In contrast, *SmMYB36* inhibits phenolic acid biosynthesis in *S. miltiorrhiza* hairy roots [26].

Although previous studies have shown that *AtMYB12* is one of the important roles of the R2R3-MYB family in plants, its functions in salt stress tolerance and phenolic acid biosynthesis have yet to be defined. In the present study, the ectopic expression of *AtMYB12* in *Salvia miltiorrhiza* revealed that *AtMYB12* can regulate the accumulation of phenolic acids and promote the response to salt stress in *Salvia miltiorrhiza.*

## 2. Results

### 2.1. Generation of Transgenic Hairy Roots and Transgenic Plants

To examine the role of AtMYB12 in transgenic *S. miltiorrhiza*, we engineered plasmids pCAMBIA1301–AtMYB12 (Appendix A), with genes under the control of the constitutive CaMV 35S promoter, and introduced them into *S. miltiorrhiza* using Agrobacterium-mediated genetic transformation. Putative transgenic lines selected on media containing Hygromycin B were confirmed using PCR with gene-specific primers (Appendix A). In total, 23 independent transgenic hairy roots and 27 transgenic plants were obtained with a transformation rate of 0.50 and 0.45, respectively. All the obtained transgenic hairy roots and transgenic plants showed a normal phenotype, and there was no significant difference in morphology.

### 2.2. Transcript Abundance of AtMYB12

To determine the transcript level of *AtMYB12*, ten transgenic hairy roots and ten transgenic plants were selected from all the identified positive lines for RNA extraction and reversely transcribed into cDNA, respectively. Expression profiles were detected via qRT-PCR analysis, and different levels of *AtMYB12* were detected in the ten transgenic hairy roots (Figure 1A) and ten transgenic plants (Figure 1B). These results indicated that *AtMYB12* was introduced into *S. miltiorrhiza* hairy roots and *S. miltiorrhiza* plants but with different expression levels. Three transgenic hairy root lines (Line 4, 6, and 8) and three transgenic plant lines (Line 2, 3, and 4) with higher expression levels of *AtMYB12* were selected for further study.

### 2.3. Overexpression of AtMYB12 Significantly Increased Salt Tolerance in Transgenic S. miltiorrhiza

To determine whether or not the expression of *AtMYB12* was associated with salt tolerance in *S. miltiorrhiza*, short-term and long-term salt stress experiments were carried out with detached leaves and whole-plant seedlings, respectively. In the short-term salt stress experiments, the detached leaf discs of three transgenic lines, *OE-AtMYB12-L2/L3/L4* exhibited significantly enhanced tolerance to 4 days of 100 mM and 250 mM NaCl treatment (Appendix A). In the long-term salt stress experiments, WT, empty vector control pCAMBIA1301, and three transgenic lines were subjected to 15 days of salt stress (Figure 2). We noticed that in the early stage of salt treatment (3 days after treatment, 3 DAT), the leaves of WT and pCAMBIA1301 turned yellow, while no significant changes were observed in transgenic lines; at 7 DAT, the leaves of WT and pCAMBIA1301 were severely damaged and necrotic, while the transgenic lines were more green and wilted less, although their leaves were also harmed. Certainly, with the continuation of salt treatment, all test lines of *S. miltiorrhiza* seedlings were dead at 15 DAT. No significant difference was observed under control conditions (Figure 2). The above results indicated that *AtMYB12* transgenic *S. miltiorrhiza* seedlings were more resistant to salt stress than were WT and pCAMBIA1301 seedlings.

### 2.4. Overexpression of AtMYB12 Reduced ROS Accumulation under Salt Stress in Transgenic S. miltiorrhiza

Salt stress increases ROS production in plants. To determine whether or not transgenic modifications could reduce ROS accumulation under salt stress, nitroblue tetrazolium (NBT) and diaminobenzidine (DAB) staining were used to examine the accumulation of O_2_^−^ and H_2_O_2_ in this study. The results showed that almost no difference was observed between WT, pCAMBIA1301, and transgenic *S. miltiorrhiza* under normal conditions, but transgenic *S. miltiorrhiza* leaves showed lower staining intensities compared to those of the WT under the treatment with 250 mM NaCl, which indicated that the overexpression of *AtMYB12* significantly reduced salt-induced ROS production in transgenic *S. miltiorrhiza* (Figure 3A,B). The determination of O_2_^−^ and H_2_O_2_ content showed that ROS accumulation in transgenic lines was significantly less than that in the WT, with a 48.01–68.14% decrease for O_2_^−^ and a 27.46–37.69% decrease for H_2_O_2_ (Figure 3C,D).

### 2.5. Overexpression of AtMYB12 Changed the Contents of EL and Chl in Transgenic S. miltiorrhiza

Electrolyte leakage (EL) and total chlorophyll content were calculated under 7 days of salt stress. Compared with the WT and the empty vector control pCAMBIA1301, all *AtMYB12* transgenic lines had a significantly lower EL, with line *OE-AtMYB12-L4* being nearly half as low as that of the WT (Figure 4A). The *AtMYB12* transgenic lines showed significantly higher total chlorophyll contents, wherein that of line *OE-AtMYB12-L4* increased 55.64% more than did that of the WT (Figure 4B).

### 2.6. Effect of AtMYB12 Expression on MDA Content and SOD, POD, and CAT Activities

To determine whether or not the transgenic plants could mitigate cellular damage and ROS accumulation caused by salt, we measured the physiological indices of WT, empty vector control pCAMBIA1301, and *AtMYB12* transgenic lines after 7 days of salt treatment; MDA concentrations were significantly lower in all transgenic lines than they were in the WT and pCAMBIA1301, which demonstrated that the overexpression of *AtMYB12* could have alleviated cell membrane damage in transgenic plants (Figure 5A). Salt stress responses were further analyzed by determining SOD, POD, and CAT activities, which are crucial for ROS scavenging and protecting the cells and the organism from oxidative damage under stress. The enzymatic activities of SOD, POD, and CAT were exhibited as much higher in all *AtMYB12* transgenic lines than they were in the WT with the range 21.07–78.03% (Figure 5B–D). These results indicated that the overexpression of *AtMYB12* significantly increased salt tolerance in transgenic *S. miltiorrhiza* via decreased MDA content and enhanced antioxidant enzyme activity.

### 2.7. Transcriptomic Analysis of AtMYB12 Transgenic S. miltiorrhiza

To further investigate the molecular mechanisms underlying the enhanced salt tolerance in *AtMYB12* transgenic *S. miltiorrhiza*, we compared the global expression profiles of *AtMYB12* transgenic lines versus those of WT plants. The quality inspection data are given in Appendix A. In total, 510 differentially expressed genes (DEGs) were identified in *AtMYB12* transgenic lines versus WT, including 179 up-regulated genes and 331 down-regulated genes (Appendix A). According to the classification of the KEGG annotations of DEGs, we noticed that many genes were involved in plant hormone signal transduction and the MAPK signaling pathway, both belonging to the category of ‘environmental information processing’ (Appendix A), indicating that *AtMYB12* may enhance the salt resistance of transgenic *S. miltiorrhiza* by regulating gene expression levels involved in responding to changes in the outside environment.

### 2.8. Overexpression of the AtMYB12 Promotes Salvianolic Acid Accumulation

To investigate whether or not the active ingredients were changed via *AtMYB12* overexpression, the contents of water-soluble salvianolic acid B and lipid-soluble tanshinones (tanshinone I, T-I; tanshinone IIA, T-IIA; cryptotanshinone, CT) were measured in the transgenic hairy roots and transgenic plants, respectively. Although no distinct differences in phenotype were apparent between all tested lines (Figure 6), high-performance liquid chromatography (HPLC) analysis revealed that the salvianolic acid B concentrations in the three transgenic hairy root lines and three transgenic plant lines were all significantly increased compared with those in the WT (Figure 7). *OE-AtMYB12-L8* was the best transgenic hairy root line with the content of 33.99 mg/g dw, which was 2.04-fold that in the WT with the content of 16.64 mg/g dw (Figure 7A). Moreover, the best transgenic plant line, *OE-AtMYB12-L3*, yielded 1.56-fold higher levels of salvianolic acid B, in an amount of 45.59 mg/g dw, compared with the WT (Figure 7B). However, no significant changes were found in the content of three tanshinones (T-I, T-IIA, and CT) in all the tested lines (Appendix A).

To determine whether or not the elevated accumulation of salvianolic acid in the *AtMYB12* transgenic lines was a result of up-regulated gene expression, the transcript levels of seven genes related to salvianolic acid biosynthesis in *AtMYB12* transgenic hairy roots and transgenic plants were analyzed via qRT-PCR. The qRT-PCR results indicated that most of these genes, including *SmPAL1*, *SmC4H1*, *Sm4CL1*, *SmTAT1*, and *SmHPPR1*, were significantly up-regulated in transgenic hairy roots (Figure 8A) and transgenic plants (Figure 8B), particularly the expression level of *SmC4H1*, which showed the largest fold change in both transgenic hairy roots and transgenic plants. All these results suggested that *AtMYB12* exhibits a promotion effect on phenolic acid biosynthetic genes.

## 3. Discussion

*Arabidopsis MYB12* is one of the R2R3-MYB family members which participates in various aspects in planta. Previous research has indicated that transgenic tobacco-carrying *AtMYB12* enhanced plant resistance to aphid-infested (*Aphidoidea*) pests [27]. Moreover, it was reported that the overexpression of *AtMYB12* enhanced salt and drought tolerance in transgenic *Arabidopsis* [28]. In our study, we demonstrated the function of *AtMYB12* through both the short-term and long-term salt stress experiments; leaves of *AtMYB12* transgenic lines suffered much less damage than did wild-type plants, and showed a more resistant phenotype under high-salinity conditions (Figure 2 and Appendix A). Consequently, we reached the same conclusion that the overexpression of *AtMYB12* effectively improved tolerance to salt stress in *S. miltiorrhiza*.

Cellular ROS (reactive oxygen species) homoeostasis is critical for plant growth, development, and responses to adverse conditions [29]. The production of ROS, such as superoxide anion (O_2_^−^), hydrogen peroxide (H_2_O_2_), and their more toxic byproducts, singlet oxygen (^1^O_2_) and hydroxyl radicals (OH^−^), will increase when plants are exposed to various biotic and abiotic stresses [30]. Salt stress destroys the balance between ROS generation and scavenging, resulting in ROS over-accumulation, causing gigantic damage to plants. At this point, the activity of antioxidant enzymes in plants also increases to alleviate the harm caused by salt stress [31]. For instance, *SlMAPK3* enhanced salt tolerance in tomato plants by scavenging ROS accumulation and up-regulating the expression of ethylene-signaling-related genes [32]. The overexpression of *PvNAC1* in switchgrass (*Panicum virgatum* L.) enhanced tolerance to salt stress through higher ROS scavenging ability and less Na^+^ and more K^+^ accumulation in roots and shoots [33]. Our studies confirmed that *AtMYB12* transgenic *S. miltiorrhiza* plants displayed higher SOD, CAT, and POD activities than those of the WT, accompanied by lower MDA levels and O_2_^−^ and H_2_O_2_ content (Figure 3 and Figure 5). These results indicated that overexpression of *AtMYB12* can reduce the degree of damage to the plant cell membrane lipid structure under salt stress, and enhance the activity of ROS scavenging enzymes in order to alleviate the adverse effects of salt stress on plants. The above results are consistent with those of the previous research on the relationship between ROS homeostasis and plant salt resistance [34,35].

Except for the above-mentioned analysis, the transcriptome analysis of *AtMYB12* transgenic lines versus WT plants suggested that *AtMYB12* may improve salt tolerance by regulating the gene expression levels in response to environmental stress in *S. miltiorrhiza*, because many differentially expressed genes could participate in hormone signal transduction and the MAPK signaling pathway (Appendix A). In the later stage, we will perform transcriptome sequencing on transgenic plants before and after salt stress to conduct more in-depth molecular mechanism research on improved salt tolerance in *AtMYB12* transgenic *S. miltiorrhiza*.

In terms of participating in plant physiological metabolic processes, *AtMYB12* was initially considered a flavonol-specific transcription factor in *Arabidopsis thaliana*. In *Arabidopsis*, *AtMYB12* up-regulated the expression of chalcone synthase (CHS) and flavonol synthase (FLS) genes, resulting in the accumulation of flavonoids [36]. In tomatoes, there was an increase in the content of different flavonoids in leaves of the transgenic lines expressing *AtMYB12* [37]. However, Meng et al. found that flavonoid accumulation was at the expense of the great sacrifice of L-phenylalanine in genetic engineering (GE) tomatoes by using the proposed new multi-omics method for GE plant evaluation [38]. Meanwhile, the overexpression of *AtMYB12* led to the production of rutin in transgenic tobacco callus culture [39] and buckwheat hairy root cultures [40]. Furthermore, the *OE-AtMYB12* transgenic tobacco lines exhibited stronger resistance to pests because of the ameliorated content of rutin [41].

Our study demonstrated that *AtMYB12* is a positive regulator of phenolic acid biosynthesis in *S. miltiorrhiza*. We determined the concentration of salvianolic acid B (SalB) via HPLC; the SalB concentration increased to 33.99 mg/g dw and 45.59 mg/g dw in *AtMYB12* transgenic hairy root Line 8 and transgenic plant Line 3, respectively, which was 2.04 times and 1.56 times that of the wild type (Figure 7). Furthermore, Zhang et al. found that *AtPAP1* plays an important role in the accumulation of phenolic acids in *S. miltiorrhiza*, and the high-SalB phenotype is stable during different developmental stages in the roots [42]. According to the results of qRT-PCR analysis, most genes involved in the phenolic acid biosynthetic pathway were significantly up-regulated in *AtMYB12* transgenic *S. miltiorrhiza* (Figure 8). Based on these, we suggest that *AtMYB12* may promote the accumulation of phenolic acids by upregulating the expression levels of genes involved in the biosynthetic pathways, one of which, the phenylpropanoid-derived pathway, is a general pathway for the synthesis of phenolic acids and flavonoids [43]. Previous studies have shown that an overexpression of *AtMYB12* in tomatoes induces multiple genes in the flavonoid biosynthesis pathway [44]; therefore, we speculate that the gene expression patterns are also important part of *AtMYB12* regulating phenolic acid biosynthesis in *S. miltiorrhiza*. Further experiments are needed to verify the supposition in the future.

In the present study, *AtMYB12* transgenic *S. miltiorrhiza* exhibited improved salt resistance in the face of environmental stress. The tolerance to high salinity has important reference significance for the development and utilization of germplasm resources of *S. miltiorrhiza*, as well as for planting and growth in adverse environments [45,46]. Our findings demonstrate that *AtMYB12* is effective in promoting the accumulation of phenolic acids, especially SalB, in *S. miltiorrhiza*. Although the molecular regulatory mechanism of *AtMYB12* in phenolic acid biosynthesis is unclear, our study demonstrates that it can significantly regulate this biosynthesis process at least. Therefore, the study provided a good foundation for elucidating the molecular mechanism regulated by *AtMYB12*; meanwhile, it has positive significance for the molecular breeding of high-grade *S. miltiorrhiza*. In terms of human nutrition and health, the high-SalB-concentration lines of *AtMYB12* transgenic *S. miltiorrhiza* could be tested as high-quality medicinal sources for the treatment of cardiovascular and cerebrovascular disease.

## 4. Materials and Methods

### 4.1. Vector Construction 

The full-length ORF of *AtMYB12* (NCBI reference sequence AT2G47460) was amplified with the designed specific primers, AtMYB12-F/R, containing the restriction site *Bgl* II/*Bst* EII. Polymerase chain reaction (PCR) amplification was performed using PrimeSTAR Max DNA polymerase (Takara, Dalian, China). The *AtMYB12* gene was amplified and integrated into the binary vector pCAMBIA1301 using SE Seamless Cloning and Assembly Kit (ZOMANBIO, Beijing, China). The recombinant expression vector, *pCAMBIA1301-AtMYB12*, was sequenced using Sanger technology (Biotech, Shanghai, China). Hygromycin resistance was used as a selectable marker. The empty vector and recombinant vector were transformed individually into the *Agrobacterium tumefaciens* strain LBA4404 and *Agrobacterium rhizogenes* strain C58C1 (pRiA4) by using standard heat-shock methods for plant transformation experiments [47]. All primers used in the present study are listed in Appendix A.

### 4.2. Plant Materials, Genetic Transformation, and Molecular Characterization

*S. miltiorrhiza* plants were grown in hormone-free Murashige and Skoog (MS) medium containing 3% sucrose and 0.8% agar (pH 5.8 ± 0.1), at 25 ± 2 °C under 16 h a light/8 h dark photoperiod [47,48]. Transgenic hairy roots and transgenic plants were obtained via the improved Agrobacterium-mediated leaf disk method, as described previously [49,50].

Genomic DNA from Hygromycin-resistant hairy roots was extracted using a modified CTAB (cetyltrimethylammonium bromide) method [48,49]. Specific primers 35S-F (based on the sequences of the CaMV 35S promoter, for the forward primer) and *AtMYB12*-R (for the reverse primer) were used to amplify the DNA template, together with the detection of the *rolB* gene. Obtained transgenic hairy roots were further cultured in a flask with a volume of liquid 1/2 B5 medium (100 mL) in the dark at 25 °C on an orbital shaker set at 120 rpm for two months.

Meanwhile, leaves of seedlings grown in rooting medium (containing 30 mg/L Hygromycin B) were sampled. The same CTAB method was used to isolate genomic DNA from these different potential transgenic plants. The above samples were identified via PCR using the primer pairs 35S-F/*AtMYB12*-R and Hyg-F/R. Positive transgenic plants were then grown in soil for further observation and stress treatment.

### 4.3. RNA Isolation and Detection of Gene Transcription Level 

For the exploration of the transcription level of *AtMYB12* and related biosynthetic pathway genes, total RNA was extracted from wild-type (WT) and transgenic hairy roots/transgenic plants using the Easy RNA extraction kit (Promega, Beijing, China). Gene expression levels were measured via qRT-PCR relative to the reference gene *SmActin* using the 2^−ΔΔCT^ calculation method [51,52]. All primers used for qRT-PCR analyses are listed in Appendix A.

### 4.4. Salt Stress Treatment

To analyze salt tolerance in transgenic *S. miltiorrhiza* plants, seedlings of the wild type, the empty vector control pCAMBIA1301, and the three transgenic lines with high expression were transferred to a solid MS medium for 25–30 days. The tube plantlets were transferred to flower pots (diameter 8.4 cm) containing a mixture of sterile nutritive soil and vermiculite (3:1), then cultured in a growth chamber at 25 ± 2 °C under a 16 h/8 h photoperiod for 30 days.

For short-term salt treatment, leaf discs (diameter 10 mm) were punched from the third youngest leaves with a hole punch and immediately transferred to Petri dishes (diameter 6 cm) containing 0 mM, 100 mM, and 250 mM NaCl solutions. The phenotype was observed and photographed on the fourth day after treatment. For the long-term salt treatment, the experimental group was irrigated with 250 mM NaCl solution each time, while the control group was watered with the same amount of water (1 L). Each group was irrigated once every 3 days until the seedlings died on day 15. On day 37, plants were stressed with 250 mM NaCl for 7 days. At this time, electrolyte leakage and other physiological indicators in leaves were measured on the third youngest leaf of at least three randomly selected plants per treatment. Each experiment contained three individual *S. miltiorrhiza* plants from each line, and three biological replicates were performed.

### 4.5. Determination of Physiological Indices

The accumulation of O_2_^−^ and H_2_O_2_ was observed using diaminobenzidine (DAB) and nitroblue tetrazolium (NBT), respectively. The O_2_^−^ production rate and H_2_O_2_ content were measured using a reagent kit (Beijing Solarbio Science and Technology, China) [48,49]. Electrolyte leakage (EL) was determined as previously described with some modifications [53]. In brief, 0.3 g of fresh leaf samples was placed in test tubes containing 10 mL of distilled deionized water at 40 °C for 30 min. After 30 min, the initial electrical conductivity of the medium (C1) was measured using an electrical conductivity meter. Subsequently, samples were placed in a boiling water bath (100 °C) for 20 min then cooled to 25 °C, and final electrical conductivity (C2) was recorded. The EL rate was calculated as follows: EL (%) = C1/C2 × 100.

Total chlorophyll contents from WT and transgenic plants were determined in accordance with the method of Richardson et al. The amount of total chlorophyll was calculated using the following formula: Total chlorophyll content = 6.63 × OD_665_ + 18.08 × OD_649_ [54]. Malondialdehyde (MDA) content was assayed via the thiobarbituric acid (TBA) method. The amount of MDA was calculated using the following formula: MDA content = 6.45 × (OD_532_ − OD_600_) − 0.56 × OD_450_ [55]. For the measurement of SOD, POD, and CAT activities, 0.5 g of fresh leaf samples was ground into a powder using a mortar with 5 mL of phosphate-buffered saline (pH 7.8). The supernatant (enzyme solution) was poured into a tube after centrifugation and then stored at 4 °C until use. Superoxide dismutase (SOD) activity, peroxidase (POD) activity, and catalase (CAT) activity were measured as described previously [47,48,56]. 

### 4.6. Transcriptomic Analysis of AtMYB12 Transgenic S. miltiorrhiza

In order to study the role of *AtMYB12* in the mechanism of salt resistance in transgenic *S. miltorrhiza*, empty vector control pCAMBIA1301 and *pCAMBIA1301-AtMYB12* transgenic line 4 were subjected to transcriptomic analysis. RNA concentration and purity were measured using NanoDrop 2000 (Thermo Fisher Scientific, Wilmington, DE, USA). RNA integrity was assessed using RNA Nano 6000 Assay Kit of the Agilent Bioanalyzer 2100 system. Sequencing was performed by Biomarker Technologies Co., Ltd. (Beijing, China) and a cDNA library was constructed and subjected to the Illumina NovaSeq6000 platform. The criteria for differentially expressed genes (DEGs) were set as a fold change (FC) of ≥1 and *p* value < 0.01. Pathway analysis was used to determine the significant pathway of the DEGs in accordance with KEGG (http://www.genome.ad.jp/kegg/ (accessed on 8 August 2023)) [57].

### 4.7. Determination of Salvianolic Acids and Tanshinones

To test whether or not *AtMYB12* affects the accumulation of active ingredients in hairy roots and transgenic *S. miltiorrhiza* plants, hairy roots grown for 60 days and *S. miltiorrhiza* roots harvested after growth in the experimental field for 5 months were used to determine the contents of tanshinone and salvianolic acid. The above sample materials were dried to a constant weight in an oven at 50 °C and then ground into a powder. Each sample powder (0.15 g) was extracted for 30 min with the pure methanol (25 mL) and 80% methanol aqueous solution (50 mL) under ultrasonic treatment (42 kHz) for tanshinone and salvianolic acid, respectively, and the resulting mixtures were centrifuged at 8000× *g* for 10 min. The supernatants were filtered through a 0.22 μm organic membrane filter and analyzed via HPLC.

The contents of tanshinone and salvianolic acid were determined on a EClassical 3100 HPLC system (Elite, Dalian, China) equipped with a PDA detector. Chromatographic separation was performed using a Supersil ODS2-C18 column (4.6 mm × 250 mm, 5 μm particle size) at 25 °C. The sample injection volume was 10 μL and the wavelengths used for the detection of tanshinone and salvianolic acid were 270 and 286 nm, respectively. For the tanshinone test, separation was achieved via elution using a gradient with solvents A (acetonitrile) and B (0.02% phosphoric acid solution). The gradient (*v*/*v*) was as follows: 0–6 min, 61% A; 6–20 min, 61–90% A; 20–20.5 min, 90–61% A; 20.5–25 min, 61% A. For the salvianolic acid test, separation was achieved via isotropic elution with 22% solvents A (acetonitrile) and 78% B (0.1% phosphoric acid solution).

### 4.8. Statistical Analysis

Bars represented the mean ± SD of three independent experiments. Student’s *t*-test was used to compare significant differences. * and ** represent significant differences at *p* < 0.05 and *p* < 0.01.

## Figures and Tables

**Figure 1 ijms-24-15506-f001:**
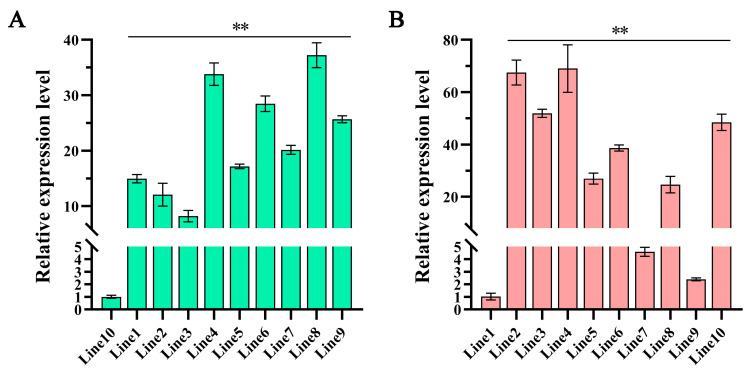
Expression pattern of *AtMYB12* in *S. miltiorrhiza* transgenic hairy roots (**A**) and *S. miltiorrhiza* transgenic plants (**B**). Transcription abundance of *AtMYB12* was normalized to the *SmActin* gene via the 2^−ΔΔCT^ method. ** represents significant differences at the *p* < 0.01 level.

**Figure 2 ijms-24-15506-f002:**
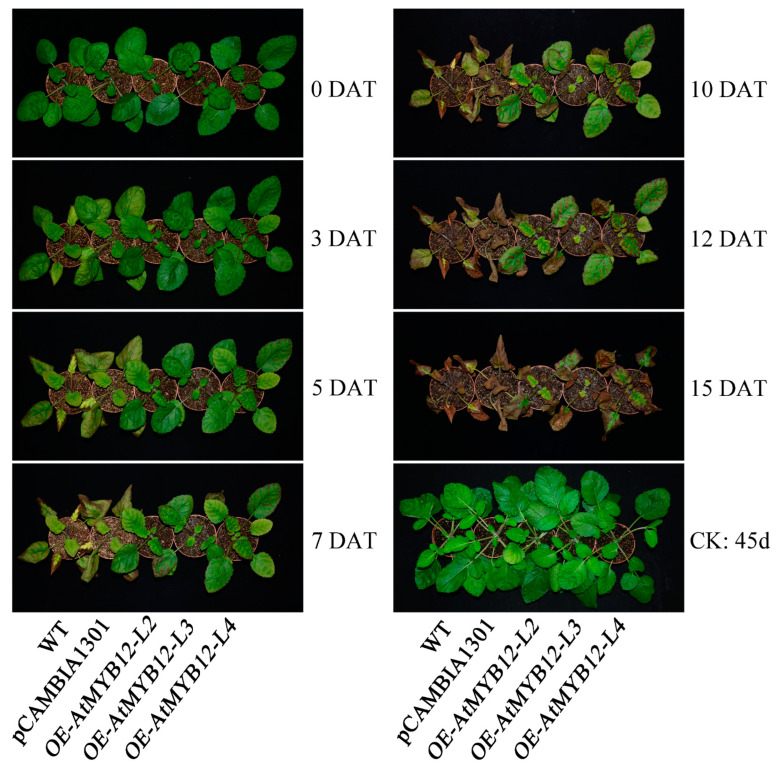
Morphological changes in WT and transgenic lines under salt stress. CK, control group irrigated with water; WT, wild type; pCAMBIA1301, empty vector control pCAMBIA1301 line; *OE-AtMYB12-L2/L3/L4* and *pCAMBIA1301-AtMYB12* transgenic lines 2/3/4; DAT, days after treatment.

**Figure 3 ijms-24-15506-f003:**
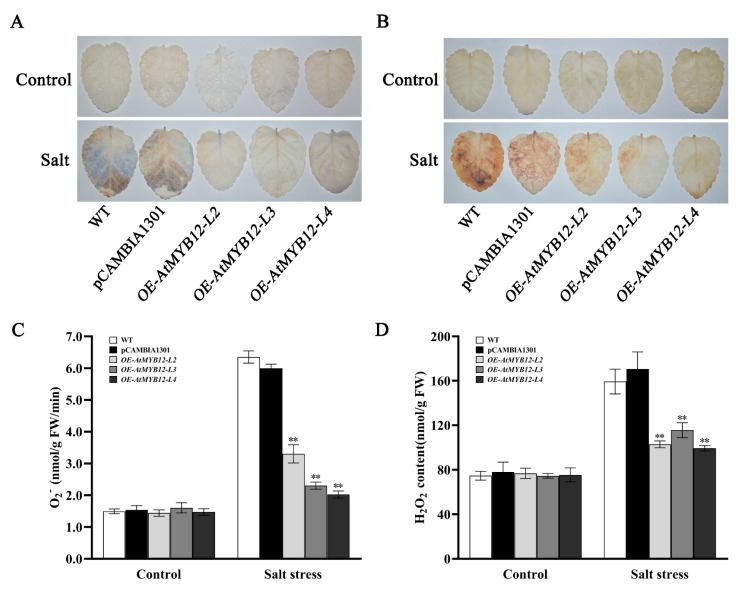
O_2_^−^ (**A**,**C**) and H_2_O_2_ (**B**,**D**) accumulation in WT and transgenic lines under salt stress. Leaves were prepared after salt treatment and incubated in NBT or DAB solution. The location and accumulation of O_2_^−^ (**A**) and H_2_O_2_ (**B**) were indicated via blue and brown staining. The O_2_^−^ production rate (**C**) and H_2_O_2_ content (**D**) were determined after salt treatment. Control, plants growing under normal conditions; Salt, plants growing under salt stress treatment (7 days). ** represents significant differences at the *p* < 0.01 level.

**Figure 4 ijms-24-15506-f004:**
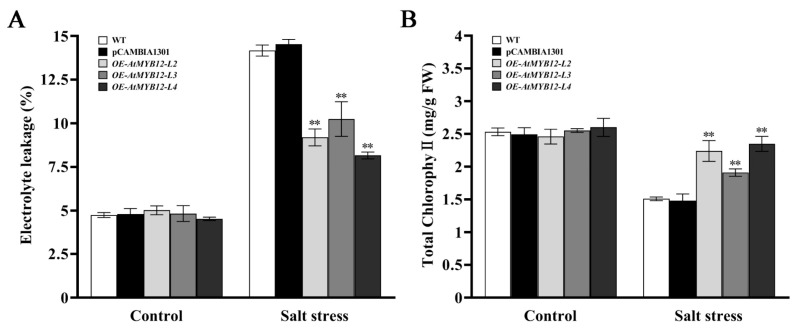
Effects of salt stress on electrolyte leakage (EL) and total chlorophyll content in WT and transgenic lines after salt treatment for 7 days. (**A**) Electrolyte leakage (EL); (**B**) total chlorophyll content. ** represents significant differences at the *p* < 0.01 level.

**Figure 5 ijms-24-15506-f005:**
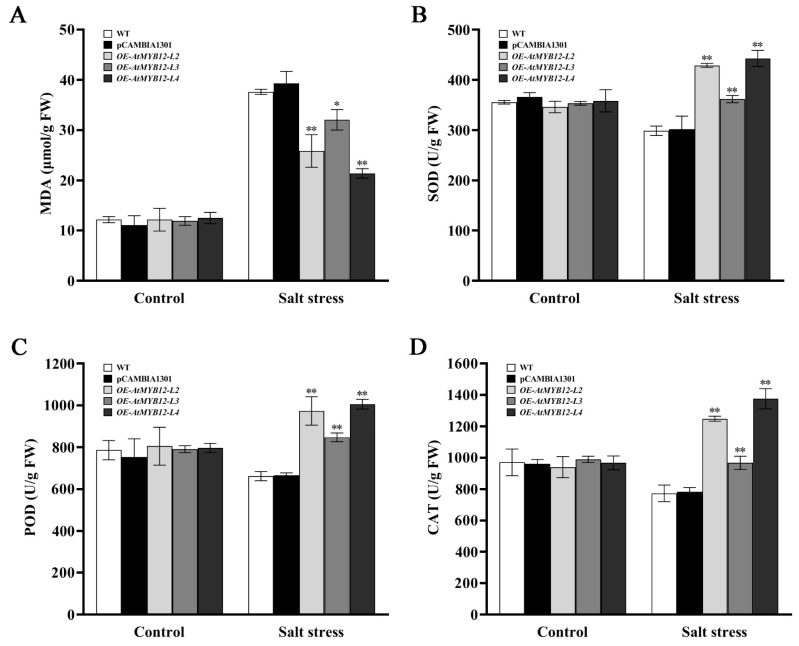
Comparison of MDA content (**A**) and SOD (**B**), POD (**C**), and CAT (**D**) activities in WT and transgenic lines after salt treatment for 7 days. * and ** represent significant differences at *p* < 0.05 and *p* < 0.01.

**Figure 6 ijms-24-15506-f006:**
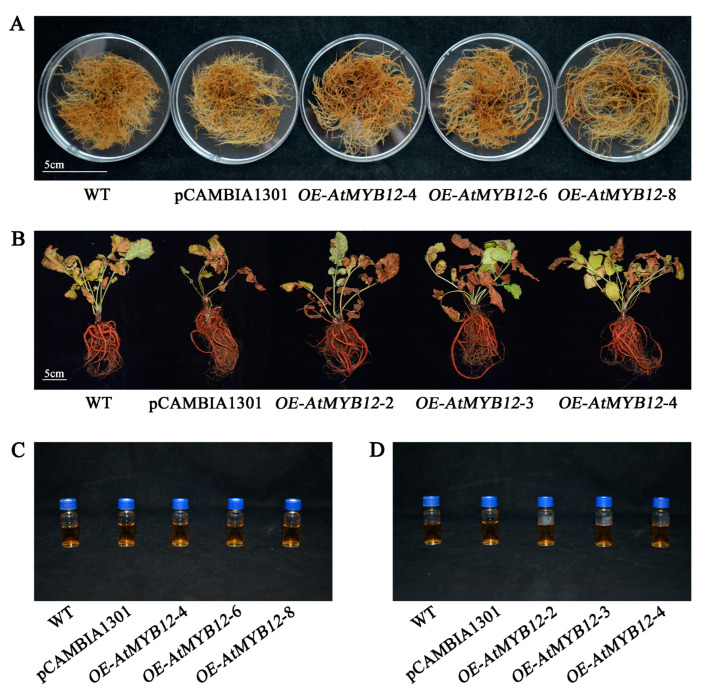
The phenotype of *S. miltiorrhiza* hairy roots/*S. miltiorrhiza* plants (scale bar = 5 cm). (**A**) *AtMYB12* transgenic hairy root lines after being cultured in 1/2 B5 liquid medium for 60 days before being photographed. (**B**) *AtMYB12* transgenic plant lines after growth in the experimental field for 5 months. (**C**) The salvianolic acid B extracts of the *AtMYB12* transgenic hairy root lines. (**D**) The salvianolic acid B extracts of the *AtMYB12* transgenic plant lines.

**Figure 7 ijms-24-15506-f007:**
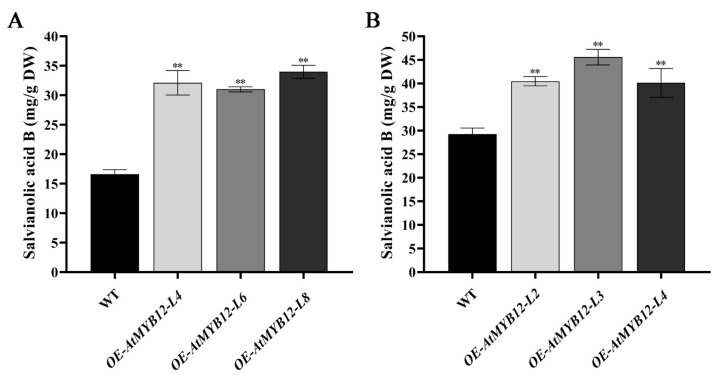
Content of salvianolic acid B in *S. miltiorrhiza* transgenic hairy roots (**A**) and *S. miltiorrhiza* transgenic plants (**B**). ** represents significant differences at the *p* < 0.01 level.

**Figure 8 ijms-24-15506-f008:**
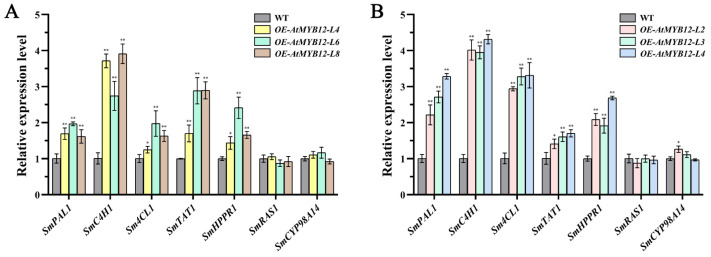
Transcript level of salvianolic acid biosynthesis pathway genes in *S. miltiorrhiza* transgenic hairy roots (**A**) and *S. miltiorrhiza* transgenic plants (**B**). * and ** represent significant differences at *p* < 0.05 and *p* < 0.01.

## Data Availability

The original data of this present study are available from the corresponding authors.

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
