# Peer review of "Simultaneous Promotion of Salt Tolerance and Phenolic Acid Biosynthesis in Salvia miltiorrhiza via Overexpression of Arabidopsis MYB12"

_ijms, 2023, doi:10.3390/ijms242115506_

Round 1

Reviewer 1 Report

First, I would like to give my sincere congratulations to the authors of the manuscript “Simultaneous Promotion of Salt Tolerance and Phenolic Acid Biosynthesis in Salvia miltiorrhiza by Overexpressing Arabidopsis MYB12". The authors present a well-written and structured manuscript. It is recommended for publication after minor proofreading for grammar and improved figure quality

Author Response

   Thank you very much for the comments from you about our manuscript submitted to International Journal of Molecular Sciences. The manuscript entitled “Simultaneous Promotion of Salt Tolerance and Phenolic Acid Biosynthesis in Salvia miltiorrhiza by Overexpressing Arabidopsis MYB12” (ijms-2617900) has been revised in detail according to the comments. The corrections have been done and indicated in red color in the revised manuscript. Thanks again for your helpful suggestion for grammar. All the comments have been incorporated into the revised manuscript.

   Attached are the revised manuscript. Hopefully, we have addressed all of your concerns.

Reviewer 2 Report

The manuscript deals with the assessment of transgenic Salvia miltiorrhiza response to salt tolerance caused by the overexpression of AtMYB12 transcription factor. The study is a nice combination of genetic and chemical analysis. However, there are some flaws that require corrections. The description of statistical analysis is missing in the Materials and methods. The results are often descriptive and do not include the values or % changes between different plants. Other comments are listed below:

L15-19: add some % changes of examined parameters between wild type and transgenic plants

L43-44: highlight the ubiquitous role of antioxidant enzymes and nonenzymatic antioxidants in alleviating different types of abiotic stresses (pesticides, salinity, drought etc.).  For this purpose the Authors may refer to the following reference: https://doi.org/10.1016/j.scienta.2022.110988

L94: indicate letters of statistical significance in the Fig. 1

L126: ‘after salt stress’ – replace after by ‘under’ or ‘exposed to’, check throughout the paper

L361: indicate pot dimensions, weight of soil and physicochemical composition of soil

L364-373: how many repetitions were used to salt treatment and determination of parameters below

L369: indicate the amount of water

L375: describe briefly determination of O2-, H2O2, chlorophyll and MDA

L406: describe the extraction and determination procedure including chromatographic conditions, column, phase composition etc.

Author Response

   Thank you very much for the comments from you about our manuscript submitted to International Journal of Molecular Sciences. The manuscript entitled “Simultaneous Promotion of Salt Tolerance and Phenolic Acid Biosynthesis in Salvia miltiorrhiza by Overexpressing Arabidopsis MYB12” (ijms-2617900) has been revised in detail according to the comments. The corrections have been done and indicated in red color in the revised manuscript. Thanks again for your helpful suggestion. All the comments have been incorporated into the revised manuscript. The responses (in BOLD type) to the comments are stated below:

1. The description of statistical analysis is missing in the Materials and methods.

Response: Statistical analysis has been added to Materials and methods.

2. L15-19: add some % changes of examined parameters between wild type and transgenic plants

Response: Thank you for your useful suggestion. The % changes of examined parameters between wild type and transgenic plants are presented in results (results 2.5 and 2.6). The corrections have been done and indicated in red color in the revised manuscript.

3. L43-44: highlight the ubiquitous role of antioxidant enzymes and nonenzymatic antioxidants in alleviating different types of abiotic stresses (pesticides, salinity, drought etc.). For this purpose the Authors may refer to the following reference: https://doi.org/10.1016/j.scienta.2022.110988.

Response: Thank you for your helpful suggestion. We have carefully referred to the article you mentioned and have revised the second paragraph of the introduction. The corrections have been done and indicated in red color in the revised manuscript.

 4. L94: indicate letters of statistical significance in the Fig. 1.

Response: We have added the letters of statistical significance in the Fig. 1.

5. L126: ‘after salt stress’ – replace after by ‘under’ or ‘exposed to’, check throughout the paper.

Response: We have replaced ‘after salt stress’ by ‘under salt stress’ , the corrections have been done and indicated in red color in the revised manuscript.

6. L361: indicate pot dimensions, weight of soil and physicochemical composition of soil.

Response: Thank you for your useful suggestion. The diameter of pot is 8.4 cm, the weight of soil is 80 g, the physicochemical composition of soil is sterile nutritive soil and vermiculite (3:1). The corrections have been done and indicated in red color in the revised manuscript.

7. L364-373: how many repetitions were used to salt treatment and determination of parameters below.

Response: Each experiment contained three individual S. miltiorrhiza plants from each line, and three biological replicates were performed. The corrections have been done and indicated in red color in the revised manuscript.

8. L369: indicate the amount of water.

Response: The experimental group was irrigated with 250 mM NaCl solution each time, while the control group was watered with the same amount of water (1 liter). The corrections have been done and indicated in red color in the revised manuscript.

9. L375: describe briefly determination of O2-, H2O2, chlorophyll and MDA

Response:

O2- production rate and H2O2 content were measured using a reagent kit (Beijing Solarbio Science and Technology, China).

Total chlorophyll contents from WT and transgenic plants were determined according to the method of Richardson et al. The amount of total chlorophyll was calculated using the following formula: Total chlorophyll content = 6.63*OD665 + 18.08*OD649.

The malondialdehyde (MDA) content was assayed by the thiobarbituric acid (TBA) method. 0.5 g fresh leaf samples were ground into a powder using a mortar with 5 mL 0.1% trichloroacetic acid (TCA) and 5 mL 0.5% TBA. The amount of MDA was calculated using the following formula: MDA content = 6.45*(OD532 - OD600) - 0.56 *OD450.

The corrections have been done and indicated in red color in the revised manuscript.

10. L406: describe the extraction and determination procedure including chromatographic conditions, column, phase composition etc.

Response: Thank you for your helpful suggestion. The revised text is as follows:

To test whether AtMYB12 affects the accumulation of active ingredients in hairy roots and transgenic S. miltiorrhiza plants, hairy roots grown for 60 days and S. miltiorrhiza roots harvested after growth in an experimental field for 5 months were used to determine the contents of tanshinone and salvianolic acid. The above sample materials were dried to constant weight in an oven at 50 °C and then grind into powder. Each sample powder (0.15 g) was extracted for 30 min with the pure methanol (25 mL) and 80% methanol aqueous solution (50 mL) under ultrasonic treatment (42 kHz) for tanshinone and salvianolic acid, respectively, and the resulting mixtures were centrifuged at 8000 g for 10 min. The supernatants were filtered through a 0.22 μm organic membrane filter and analyzed by HPLC.

The contents of tanshinone and salvianolic acid were determined on a EClassical 3100 HPLC system (Elite, Dalian) equipped with a PDA detector. Chromatographic separation was performed using a Supersil ODS2-C18 column (4.6 mm × 250 mm, 5 μm particle size) at 25 °C. The sample injection volume was 10 μL and the wavelengths used for the detection of the tanshinone and salvianolic acid were 270 and 286 nm, respectively. For tanshinone test, separation was achieved by elution using a gradient with solvents A (acetonitrile) and B (0.02% phosphoric acid solution). The gradient (v/v) was as follows : 0–6 min, 61% A; 6–20 min, 61–90% A; 20–20.5 min, 90-61% A; 20.5–25 min, 61% A. For salvianolic acid test, separation was achieved by isotropic elution with 22% solvents A (acetonitrile) and 78% B (0.1% phosphoric acid solution).

The corrections have been done and indicated in red color in the revised manuscript.

Attached are the revised manuscript. Thanks again for your helpful suggestion. 

Reviewer 3 Report

The article „Simultaneous Promotion of Salt Tolerance and Phenolic Acid Biosynthesis in Salvia miltiorrhiza by Overexpressing Arabidopsis MYB12“ is interesting and simply written.

I make several suggestions to improve the quality of the manuscript.

In the Introduction, state the botanical affiliation of the genus Salvia.

The Discussion is written in subtitles, I suggest you write it as full text. The Discussion is an important part of the article. Please improve it by comparing the Results presented in Figures 1-8 with other research results. Please refer to all Figures 1-8 in the discussion and refer to the data in the supplementary materials.

Please write the Conclusions separately.

References are not written according to journal instructions, please revise.

Author Response

Thank you very much for the comments from you about our manuscript submitted to International Journal of Molecular Sciences. The manuscript entitled “Simultaneous Promotion of Salt Tolerance and Phenolic Acid Biosynthesis in Salvia miltiorrhiza by Overexpressing Arabidopsis MYB12” (ijms-2617900) has been revised in detail according to the comments. The corrections have been done and indicated in red color in the revised manuscript. Thanks again for your helpful suggestion. All the comments have been incorporated into the revised manuscript. The responses (in BOLD type) to the comments are stated below:

1. In the Introduction, state the botanical affiliation of the genus Salvia.

Response: Thank you for your useful suggestion. Salvia miltiorrhiza Bunge belongs to the Labiatae family. We've added its botanical affiliation to the introduction.

2 and 3. The Discussion is written in subtitles, I suggest you write it as full text. The Discussion is an important part of the article. Please improve it by comparing the Results presented in Figures 1-8 with other research results. Please refer to all Figures 1-8 in the discussion and refer to the data in the supplementary materials. Please write the Conclusions separately.

Response: Thank you for your helpful suggestion. We have removed the subtitles of the discussion and wrote it as full text. In addition, we have added some comparative discussion with the results of previous studies. Considering the universal format of the latest manuscripts published in this journal and overall structure of the manuscript, we didn’t write a conclusion in subtitles, but we have added an obvious concluding paragraph at the end of the discussion. The corrections have been done and indicated in red color in the revised manuscript.

4. References are not written according to journal instructions, please revise.

Response: Thank you for your valuable suggestion. We have revised the format of the references according to the requirements of the journal.

Attached are the revised manuscript. Thanks again for your helpful suggestion.

Round 2

Reviewer 2 Report

The Authors have significantly improved the paper. I have no more comments.